# To Be, or Notch to Be: Mediating Cell Fate from Embryogenesis to Lymphopoiesis

**DOI:** 10.3390/biom11060849

**Published:** 2021-06-07

**Authors:** Han Leng Ng, Elizabeth Quail, Mark N. Cruickshank, Daniela Ulgiati

**Affiliations:** 1Centre for Haematology, Department of Immunology and Inflammation, Faculty of Medicine, Imperial College London, Du Cane Road, London W12 0NN, UK; h.ng@imperial.ac.uk; 2School of Biomedical Sciences, The University of Western Australia, 35 Stirling Highway, Crawley, WA 6009, Australia; liz.quail@uwa.edu.au (E.Q.); mark.cruickshank@uwa.edu.au (M.N.C.); 3School of Molecular Sciences, The University of Western Australia, 35 Stirling Highway, Crawley, WA 6009, Australia

**Keywords:** Notch signaling, B cell, development, leukemia

## Abstract

Notch signaling forms an evolutionarily conserved juxtacrine pathway crucial for cellular development. Initially identified in *Drosophila* wing morphogenesis, Notch signaling has since been demonstrated to play pivotal roles in governing mammalian cellular development in a large variety of cell types. Indeed, abolishing Notch constituents in mouse models result in embryonic lethality, demonstrating that Notch signaling is critical for development and differentiation. In this review, we focus on the crucial role of Notch signaling in governing embryogenesis and differentiation of multiple progenitor cell types. Using hematopoiesis as a diverse cellular model, we highlight the role of Notch in regulating the cell fate of common lymphoid progenitors. Additionally, the influence of Notch through microenvironment interplay with lymphoid cells and how dysregulation influences disease processes is explored. Furthermore, bi-directional and lateral Notch signaling between ligand expressing source cells and target cells are investigated, indicating potentially novel therapeutic options for treatment of Notch-mediated diseases. Finally, we discuss the role of cis-inhibition in regulating Notch signaling in mammalian development.

## 1. Introduction

Development of progenitor stem cells to highly specialized cell types requires tight regulation and orchestration of multiple signals. The evolutionarily conserved Notch pathway, first identified in mutated fruit flies with notches in their wings, is also involved in development of sensory organs of the peripheral nervous system [1,2]. In mammals, Notch signaling dictates the fate of multiple cell types during development of the embryo (embryogenesis and somitogenesis), including the central nervous system (neurogenesis and gliogenesis), cardiovascular system (cardiogenesis), and pancreatic system (pancreatic organogenesis), as well as hematopoiesis [2,3]. In this review, we present studies showing a fundamental role for Notch signaling in regulating development during embryogenesis and mediating cell fate decisions of common lymphoid progenitors towards the B cell lineage. Additionally, we explore bi-directional, lateral induction and cis-inhibition of Notch signaling in the context of development and disease progression.

## 2. The Notch Signaling Pathway

Humans and most other mammals express four Notch receptors (NOTCH1-4) and five families of Notch ligand; DELTA-LIKE1 (DLL1), DELTA-LIKE3 (DLL3), DELTA-LIKE4 (DLL4), JAGGED1, and JAGGED2 [2,3] (Figure 1). 

Activation of the Notch signaling pathway requires physical interaction between a Notch receptor and its cognate ligand, which elicit cell-context dependent responses and initiate the Notch signaling pathway [4,5] (Figure 2). Several crystallographic and functional studies show that interaction between the Delta/Serrate/lag-2 (DSL) domain of the Notch ligand and the epidermal growth factor (EGF)-like domains 11 and 12 of the Notch receptor is requisite for Notch pathway activation [6,7]. Isolated protein crystals provide evidence of interaction between NOTCH1 EGF11-13 and DLL4, in an antiparallel manner [8,9].

Gordon and co-workers [10] showed through crystallographic studies that in its unbound state, NOTCH2 *lin-12*/*Notch* repeats (LNR) wrap around the heterodimerization domain (HD) to sterically inhibit S2 cleavage. Upon receptor-ligand interaction, the ligand “pulls” the receptor, thereby exposing the negative regulatory region (NRR) at a specific site for S2 cleavage [11] (Figure 2). Parks and colleagues [11] hypothesized that endocytosis of the bound receptor-ligand was crucial for the activation of the Notch pathway. This was supported by in vivo evidence generated by Nichols and co-workers [12] utilizing murine cell lines. The S2 cleavage step is the rate-limiting step for release of the Notch intracellular domain (NIC) and occurs on the surface of the Notch-activated cell [13,14] (Figure 2). Two metalloproteases of the A Disintegrin and Metalloproteases (ADAM) family are involved in S2 cleavage: ADAM10 and ADAM17/TNF-α converting enzyme (TACE) [13,15]. Knockout of either *Adam10* or *Tace* is embryonically lethal, with *Adam10* knockout embryos showing similar defects to the Notch receptor knockouts [16,17] (Table 1). Further evidence suggests the mechanism of S2 cleavage is cell context-dependent, with varying degrees of severity and different cell types impacted at various stages of development [13,15].

Sequential cleavage at the S2 site further exposes the remaining NIC for S3 cleavage (Figure 2). Presenilin-dependent γ-secretase releases NIC into the nucleus for activation of Notch target genes [18,19]. Tagami and colleagues [19] showed diversity at the S3 cleavage site, with NIC cleavage observed at varying amino acid residues. The NIC fragments produced display differing half-lives and Notch activity, suggesting specificity of S3 cleavage as a mechanism for regulating Notch activation [19]. The release of NIC and transport into the nucleus is regulated by the nuclear localization signal (NLS) and mediated by the importin transport complex [20].

Upon translocation into the nucleus, NIC binds to the sequence-specific DNA binding repressor protein, CBF1 at Notch target genes [21]. CBF1 functions as the tethering point for binding of appropriate transcription co-factors to elicit cognate regulatory responses. Notch target genes are constitutively silenced through CBF1 recruitment of co-repressors and only activated via Notch signaling upon binding of NIC to CBF1. This binding recruits co-activators, thereby de-repressing expression of Notch target genes [5,22]. Like the other components of Notch signaling, CBF1 is critical for embryogenesis. Interestingly, *Cbf1* knockout causes earlier lethality than *Notch* knockout mouse models (Table 1), possibly due to its central non-redundant function in transducing signals from all Notch receptors [23].

## 3. Knockout of Notch Signaling Components Is Embryonically Lethal

Independent studies by both Swiatek [24] and Conlon [25] showed that *Notch1* knockout was embryonic lethal (Table 1). Subsequently, Hamada and collaborators [26] generated a *Notch2* knockout model wherein they observed no viable homozygous *Notch2* mutant embryos, with developmental impairments observed by day 10.5 (Table 1). Unlike the *Notch1* knockout mouse model, no somite defects were observed in *Notch2* knockouts, suggesting these Notch receptors regulate different aspects of mouse embryogenesis.

In contrast, *Notch3* knockout mice produced viable and fertile offspring (Table 1). A double *Notch1* and *Notch3* knockout [28] showed an identical phenotype to that of Swiatek [24], indicating that Notch3 is not required for embryo viability [28] (Table 1). Nonetheless, Notch3 is essential for appropriate vascular development, with *Notch3* knockouts displaying arterial deformities through delayed maturation of vascular smooth muscle cells [27].

Similarly, *Notch4* knockout mice generated by Krebs and colleagues [29] were viable and fertile, with no visible deformities. While a double knockout of *Notch1* and *Notch4* was embryonic lethal in mice, more severe deformities (fewer somites) were observed, suggesting cooperative roles in murine embryogenesis [29] (Table 1). More recently, James and colleagues [30] found that the *Notch4* knockout mouse described by Krebs [29] expressed a truncated *Notch4* transcript that appears to retain some regulatory functions. They reported an alternative knockout mouse model, whereby the *Notch4* coding region was replaced by the *LacZ* reporter, with low reporter gene expression in the tail bud and vasculature of the embryonic head, implicating Notch4 in vascular development [30] (Table 1). Further studies are required to clarify the molecular mechanisms that underpin *Notch1* and *Notch4* genetic interactions during embryogenesis.

Unsurprisingly, Notch ligands also play a crucial role in embryogenesis. Through the seminal paper by De Angelis and colleagues [31], Dll1 was shown to play an important role in somite boundary formation (Table 1).

*Dll3* function was elucidated through a mouse model generated via a mutagenesis screen. This data showed approximately 20% embryonic lethality in homozygous *Dll3* knockouts. Disruption in somite formation resulted in skeletal anomalies in surviving adult mice, highlighting the importance of the *Dll3* gene in somitogenesis [32] (Table 1).

The *Dll4* knockout is also embryonically lethal with vascular and arterial deformities frequently observed, verifying the importance of Dll4 in vascular formation [33,34,35] (Table 1).

Normal murine development also requires expression of the Serrate-related family of ligands; Jagged1 and Jagged2. Whilst *Jagged1* knockout did not affect somitogenesis, there were abnormalities in the development of vascular networks and hemorrhaging, resulting in embryonic death [36] (Table 1). Sidow and colleagues [37] identified a mutation in the *Jagged2* gene causing syndactylism (limb digit fusion) in a mouse model, highlighting the role of *Jagged2* during skeletal development. This was verified by Jiang and co-workers [38] whose *Jagged2* knockouts displayed limb defects, craniofacial deformities, and disrupted populations of a subset of T cells within the thymus.

Taken together, these seminal papers demonstrate the importance of the Notch signaling pathway for appropriate fetal development.

## 4. Notch Signal Involvement from Hematopoietic Stem Cells to B Lymphocytes

A key feature of hematopoietic stem cells (HSCs) is their ability for self-renewal, a process which relies on interaction with the surrounding bone marrow microenvironment. A myriad of signals allows HSCs to either maintain their multipotent state, or to initiate differentiation into multiple types of blood cells. NOTCH1 and NOTCH2 are expressed on HSCs, and interact with JAGGED1, JAGGED2, DLL1, and DLL4 on surrounding stromal cells within bone marrow to maintain HSCs in a self-renewal state [39,40,41,42,43,44]. Complementary in vitro and in vivo loss-of-function experiments of Notch signaling demonstrate enhanced differentiation of HSCs into progeny cells [45,46]. Taken together, these data indicate the importance of Notch signaling as a mediator of HSC fate within the bone marrow microenvironment (Figure 3).

During hematopoiesis, HSCs differentiate into either myeloid or lymphoid lineage cells, with Notch signaling involved through lineage commitment. Figure 3 summarizes the role of Notch during HSC differentiation through to the B cell lineage.

Notch receptors are expressed during different stages of B cell development. Analysis of fetal human B cells revealed that NOTCH1 is expressed early (from pro-B to immature B cells), whilst NOTCH2 is only expressed in the late pre-B cell stages, but not in pro-B, early pre-B or immature B cells. NOTCH3 was not detected at any stage of early B cell development [47]. Notch2 is expressed throughout adult murine peripheral B cell development, with very low levels in early bone marrow B cells increasing as they differentiate to plasma cells [48].

Conditional *Notch1* knockout mouse models show defective thymus and T cell development, with production of ectopic thymic B cells [49,50]. In an alternate *Notch* knockout, these thymic B cells express immature phenotypic markers, highlighting the importance of Notch signaling in determining B and T cell fate [51]. Overexpression of NIC1 within bone marrow led to ectopic expression of T cells and defective B cell development [52]. In transduced human cord cells, overexpression of NIC4 drives an immature CD4+ CD8+ T cell phenotype in bone marrow and spleen. These cells do not differentiate into B cells, suggesting these common lymphoid progenitors adopt a T cell fate [53]. Furthermore, knockout of *Cbf1* resulted in aberrant expression of B cells, whilst T cell differentiation was diminished [54]. These studies demonstrate that Notch signaling mediated through CBF1 is crucial for lineage differentiation of common lymphoid progenitors. Interestingly, overexpression of *Hes1*, the canonical Notch target gene in mice, produces a similar inhibition of B cell development, albeit to a lesser extent than *NIC1* overexpression [55].

Recent data from our laboratory has demonstrated a role for Notch signaling in mediating transcription of the mature B cell marker, *Complement Receptor 2* (*CR2*), in pre-B cells [56]. However, the molecular mechanism of Notch signaling during the pre-B to mature B cell transition remains to be fully elucidated. This stage of B lymphopoiesis is key for regulating self-tolerance. Notch signaling plays a critical role in the development of marginal zone or follicular B cells within the spleen through expression of Notch2 [48]. *Notch2* conditional knockout mice have fewer marginal zone B cells and decreased Cr2 expression within follicular B cells [48]. Similarly, conditional *Dll1* knockout mice displayed a decrease in marginal zone B cells within the spleen [57,58]. Combined, these studies indicate that interaction between Notch2 and Dll1 within the spleen regulates the development of marginal and follicular B cell subsets.

Conditional *Cbf1* knockout mice display a loss of marginal zone B cells, with a concomitant increase in follicular B cells in the spleen [59]. Recent data has challenged the paradigm of irreversible commitment of mature B cells to the follicular or marginal zone compartments. Induction of Notch2 expression in follicular B cells mediates transdifferentiation to marginal zone B cells that are phenotypically and functionally similar to endogenous marginal zone B cells, and localize within the spleen [60]. This data supports a novel role for Notch signaling in mediating plasticity between two distinct B cell subsets.

It is well established that the mastermind-like1 (MAML1) transcription factor binds to the Notch initiation complex and is crucial for the activation of Notch target genes [61,62]. Several *Maml1*-deficient mouse models show impaired development of marginal zone B cells, similar to mice lacking *Dll1* [57,63,64].

Notch signaling also plays an essential role in the survival of germinal center B cells. Antigen activated B cells residing within the germinal center require JAGGED1-mediated Notch signaling for survival and proliferation [65]. Using a γ-secretase inhibitor, it was demonstrated that antigen-activated B cells require γ-secretase dependent Notch signaling for survival in vitro. Notch signaling has also been implicated in the activation of B cells, stimulating differentiation into plasma cells. Specifically, Notch signaling through Dll1, together with the B cell receptor and CD40, is required for the activation of murine follicular B cells to proliferate and differentiation into plasma cells, resulting in antibody isotype switching [66]. Moreover, conditional expression of a dominant-negative form of Maml1 (DNMAML1) to inhibit Notch activation, resulted in reduced marginal zone B cells, phenotypically similar to conditional knockout of *Cbf1* or *Notch2* [48,59,66]. Following stimulation by lipopolysaccharides, splenic B cells isolated from a Notch1 knock down mouse model showed reduced antibody production [67]. Using ex vivo co-culture of murine follicular B cells expressing DNMAML1 with stromal cells expressing Dll1, a decrease in terminal differentiation of plasma cells was observed. Additionally, Notch1 is required to activate the Notch pathway for differentiation of plasma cells [68]. Combined, these data clearly demonstrate the importance of Notch in the activation, proliferation, and differentiation of mature B cells into plasma cells (Figure 3).

B1 B cells are another subset of cells that are regulated by Notch, specifically in the commitment to either the B1 or B2 lineage. In a haploinsufficient *Notch2* mouse model, a significant reduction in B1 B cells and marginal zone B cells was detected [69]. This supports the requirement of Notch2 for normal marginal zone B cell development. Constitutive expression of NIC2 results in ectopic Notch signaling. This signal promotes B1 B cell development but suppresses the progression of pre-B cells into B2 marginal zone cells [70]. Thus, the pre-B cells receiving ectopic Notch signal are inhibited in their development into marginal zone B cells [70].

## 5. Notch-Related Diseases

Dysregulation of Notch signaling is associated with human developmental disorders and cancers. Specific mutations identified within Notch receptors or ligands have been linked to rare congenital diseases and tumorigenesis.

### 5.1. Notch Associated Hereditary Diseases

Missense mutations in cysteine residues within any of the 34 EGF domains of the *NOTCH3* gene have been identified in the majority of Cerebral Autosomal Dominant Arteriopathy with Subcortical Infarcts and Leukoencephalopathy (CADASIL) patients [71,72]. While the exact mechanism causing CADASIL is unknown, development of vascular smooth muscle cells is impacted, with the disorder primarily manifest by cerebral white matter lesions [73]. Although insightful in understanding the pathology, mouse models with *Notch3* mutations or deletions do not fully recapitulate the clinical features of CADASIL [74].

Alagille syndrome is a rare autosomal dominant disorder resulting from haploinsufficiency of either *JAGGED1* or *NOTCH2*. Multiple organs are affected due to dysregulation of Notch signaling, particularly liver and kidney [75]. In ~90% of diagnosed cases, *JAGGED1* mutations have been identified within the extracellular domain [76,77]. In rarer cases, *NOTCH2* mutations within the EGF or ankyrin repeats have been reported [78,79].

In contrast to the haploinsufficiency observed in Alagille syndrome, Hadju–Cheney syndrome (HJCYS) results from frameshift or nonsense mutations in the *NOTCH2* gene. This causes deletion of the proline, glutamic acid, serine, and threonine (PEST) domain, resulting in gain-of-function of Notch signaling in HJCYS patients [80,81]. Features of this disorder include severe osteoporosis and craniofacial defects [82]. Similar clinical features to HJCYS are observed in serpentine fibula-polycystic kidney syndrome (SFPKS), which is further characterized by polycystic kidney disease. Mutations within the last exon of *NOTCH2* in SFPKS patients have been identified [83]. The similarities in clinical features of these disorders suggests they fall within a spectrum of variability of NOTCH2 gain-of-function heritable diseases.

Spondylocostal dysostosis is a rare skeletal disorder characterized by malformation of vertebrae and fusion of spine and ribs. The most common causative loss-of-function mutations lie within *DLL3* [84,85]. Knock out of *Dll3* causes similar skeletal defects, further verifying its importance in skeletal development. Other disease-causing gene mutations impacting Notch signaling include: *MESP2*, *HES7,* and *LFNG* [86,87,88].

Adams–Oliver syndrome (AOS) is characterized by mutations in *NOTCH1*, *CBF1,* or *DLL4*. Clinical features of this disorder include underdeveloped skull and terminal limb defects [89,90,91,92,93]. These mutations only account for <50% of AOS cases [89]. Loss-of-function of NOTCH1 is observed in AOS across the *NOTCH1* gene [90,91]. In CBF1-affected AOS patients, CBF1 binding affinity to its cognate binding sites is diminished, demonstrating the importance of CBF1 as an anchoring transcription factor mediating Notch signaling [92]. There have been no genotype–phenotype correlations identified in AOS patients with *DLL4* mutations [93].

Collectively, the various mutations identified in components of the Notch signaling pathway cause diverse developmental defects, highlighting the critical role of Notch in skeletal and organ development.

### 5.2. Notch Dysregulation in Hematological Malignancies

The first evidence of aberrant Notch signaling in oncogenesis was seen in T-acute lymphoblastic leukemia (T-ALL). Rare chromosomal translocations involving *NOTCH1* and *TCRβ* genes resulted in a truncated NOTCH1 protein and aberrant signaling [94]. More frequently (>50% of cases), mutations in hotspots within the HD and/or PEST domains of the *NOTCH1* gene have been characterized [95] (Figure 3). These mutations result in either ligand-independent aberrant activation of Notch signaling, or an increased stability of NIC1 within the nucleus. Aberrant NOTCH1 signaling results in a rapid increase of H3K27 acetylation in a distal enhancer of the *MYC* gene, thus inducing expression [96]. This elevated MYC expression is a molecular driver of T-ALL. *NOTCH3* is a putative NOTCH1 transcriptional target and increased NOTCH3 surface expression together with hyperactivation is frequently observed in T-ALL [97] (Figure 3). Indeed, increased NOTCH3 expression promotes T-ALL survival during endoplasmic reticulum stress [98]. Further, NOTCH3 expression in T-ALL is post-translationally regulated through the lysosomal pathway, providing a potential therapeutic target via histone deacetylase inhibition [99].

Constitutive activation of Notch signaling is associated with chronic lymphocytic leukemia (CLL), with increased expression of NOTCH1, NOTCH2, JAGGED1, and JAGGED2 [100] (Figure 3). Supraphysiological Notch signaling results in apoptosis resistance through increased expression of the anti-apoptotic protein Mcl-1. Aberrant Notch signaling reduces phosphorylation of the translational initiation factor, eIF4E, which regulates Mcl-1 [101]. Furthermore, IL-4 increases NIC1 and NIC2 expression within CLL cells, with concomitant hyperactivation of PI3K/AKT and PKCδ signaling, respectively. These signals ultimately contribute to sustained survival of CLL cells [102].

In splenic marginal zone B-cell lymphoma (SMZL), mutations within the *NOTCH2* gene have been identified in ~25% of patients (Figure 3). Most *NOTCH2* mutations in SMZL cases cluster around the transcriptional activation domain (TAD) and PEST domain, resulting in increased stability of NIC2 within the nucleus [103,104,105]. Less frequently in SMZL patients, mutations in *NOTCH1* and other components of Notch signaling have been detected. Collectively, these mutations reinforce the importance of Notch signaling within the mature B cell lineage [104]. A combination of the ligand-rich microenvironment and gain-of-function Notch activating mutations provides a selective advantage for mutant cells and predisposes these B cells to oncogenesis [106].

Notch activation in mantle cell lymphoma is driven by increased expression of MYC. Chromatin immunoprecipitation sequencing (ChIP-seq) identified a B lineage specific Notch mediated enhancer region unique to mantle cell lymphoma cell lines that is absent in T-ALL [107]. Further, Notch signaling mediates clustering of enhancer interactions with oncogenes. In mantle cell lymphoma, these three-dimensional cliques (3D cliques) are observed between B lineage specific genes and multiple enhancers [108]. Collectively, these studies suggest that targeting a combination of lineage specific oncogenes and aberrant Notch signaling could provide therapeutic treatment for specific cancer types. It is worth noting that the observed Notch mediated 3D cliques are not unique to mantle cell lymphoma. This phenomenon is also observed in Notch-addicted triple negative breast cancer cell lines with cell lineage specific oncogenes [108].

Remarkably, in a study of diffuse large B cell lymphoma (DLBCL), patients concurrently infected with Hepatitis C showed a high level of *NOTCH1* and *NOTCH2* mutations (25% of all DLBCL cases). In contrast, only a single patient (from a cohort of 64) without Hepatitis C infection had any *NOTCH* mutation [109]. *NOTCH* mutations correlated with lower overall survival, highlighting the need to focus on Notch signaling as a potential therapeutic target in DLBCL patients with Hepatitis C infection.

Whilst we have focused on aberrant Notch signaling in hematological malignancies, supraphysiological Notch signaling also contributes to solid cancers. Notch gain-of-function mutations are associated with breast cancer [110,111], glioblastoma [112,113], and adenoid cystic carcinoma [114]. Notch loss-of-function mutations are well characterized in squamous cell carcinoma [114], small cell lung cancer [114,115], and colorectal cancer [114,116].

## 6. Bi-Directional Notch Signaling

Whilst the response to Notch in target cells is well characterized, post-signaling effects on Notch ligand-expressing source cells is not well understood. Emerging evidence suggests that Notch source cells may also be transducing signals upon Notch receptor-ligand engagement, a phenomenon termed bi-directional signaling (Figure 4A). Similar to NIC cleavage in Notch target cells, Notch source cells undergo cleavage of the Notch ligand intracellular domain (ICD). In vitro assays have revealed that γ-secretase is responsible for cleaving Delta and Jagged ligands and releasing the ICD, which can be translocated into the nucleus of the Notch source cell [117,118,119]. In vitro studies show that JAGGED1-ICD (J1-ICD) reduces Notch mediated reporter activity, suggesting a role for Notch ligands in regulating the strength of the Notch signal. In these studies, J1-ICD binds to the AP1 transcriptional element, implicating the ICD of Notch ligands in regulation of gene transcription [117] (Figure 4A). Furthermore, in murine neural stem cells, Delta-like 1-ICD (D1-ICD) was shown to translocate to the nucleus upon co-culture with Notch expressing cells where it interacted with transcription factors, Smad2, Smad3, and Smad4, to stimulate TGF-β/Activin signaling [120]. Collectively, these findings suggest Notch ligand ICDs act as transcriptional co-factors within Notch source cells (Figure 4A).

Animal models also implicate a functional role for bi-directional Notch signaling. In the aforementioned Notch1 knock down mouse, lipopolysaccharides caused reduced Jagged1 expression with concomitant reduction in antibody production, which was rescued when these mutant B cells were cultured in the presence of Notch1. This mimicked splenic B cells interacting with surrounding stromal cells [67]. It is speculated that Jagged1 can regulate antibody production of activated B cells through the J1-ICD.

In a conditional *Dll1* knock out mouse model, long-term survival of activated CD4+ T cells is impacted. T cell survival was partially rescued by transducing with D1-ICD, implying Dll1 has a role in T cell survival and signaling. Interestingly, Notch signaling and expression of Notch target genes were not affected in these partially rescued T cells, suggesting that D1-ICD regulates non-canonical Notch-independent pathways [121].

Further, in a transgenic mouse model with inducible expression of J1-ICD, cardiomyocytes displayed a range of functional defects. Expression of J1-ICD was associated with Notch inhibition and activation of the Wnt/Akt pathways [122]. Notch inhibition blocks neonatal cardiomyocyte proliferation, whilst Wnt/Akt is activated as a compensatory mechanism to a lack of Notch signaling.

Collectively, many studies have shown that Notch ligand ICDs mediate transcriptional changes within the Notch ligand-expressing source cell. However, the molecular mechanism of Notch ligand ICD interaction with target genes remains to be fully elucidated. Identifying binding partners of Notch ligand ICD in the appropriate cellular context requires further investigation. While the mechanisms and impact on disease remain to be explored, the aforementioned studies suggest the role of bi-directional Notch signaling is underappreciated and could plausibly be significant in the relay of pathological signals.

## 7. Lateral Induction of Notch Signaling and Interaction with the Micro-Environment

Cancer cells can manipulate spatial and temporal expression of Notch ligands autonomously and to the surrounding stromal cells, which may modulate survival and promote malignancy. Hoare and collaborators [123] reported the dynamics of Notch signaling during Ras-oncogene-induced senescence, finding that Notch activity increases and modulates the composition of the secretome. Notch signaling switched the senescence associated secretory phenotype from a pro-inflammatory signature to a TGF-β signature. They demonstrate that during senescence, Notch signaling mediates upregulation of JAGGED1 ligand expression on neighboring cells, potentiating Notch responses in a process described as lateral induction [123] (Figure 4B). Subsequently, Parry and colleagues [124] compared senescence induced by Ras-oncogene and ectopic NIC expression. They found that Ras oncogenes and NIC triggered distinctive molecular programs resulting in differing senescence-associated secretory phenotypes. Induction of both Ras and NIC pathways demonstrated that the NIC senescence program impairs the Ras-induced pathway by blocking chromatin remodeling and heterochromatin foci. Moreover NIC-induced senescence was accompanied by increased JAGGED1 expression thereby activating Notch signaling in neighboring cells. In both studies, ectopic NIC1 induced expression of JAGGED1, suggesting a role for the lateral induction of Notch in senescence and immune surveillance via NOTCH1 and JAGGED1 [123,124]. Notch signaling between CLL and stromal cells promotes survival and chemoresistance to drug treatment. CLL cells co-cultured with bone marrow-derived mesenchymal stem cells display increased NOTCH4 and DLL3 expression. Further, in these co-cultures the mesenchymal stem cells also overexpress NOTCH4, suggestive of lateral induction of Notch signaling [125].

Cytokine expression has been shown to promote Notch signaling in hematological malignancies and enhance cellular survival within the tumor and neighboring cells. Multiple myeloma is a well characterized plasma cell neoplasm. Within the bone marrow, multiple myeloma cells develop IL-6 independence from the surrounding stromal cells in a cell autonomous manner through Notch activation of the *IL-6* gene. Further, aberrant JAGGED1 surface expression on multiple myeloma cells induces IL-6 secretion from surrounding bone marrow stromal cells, thus activating Notch [126]. Inhibition of Notch signals in either multiple myeloma cells or stromal cells reduces multiple myeloma proliferation in vitro [126,127]. In CLL, IL-4 induces JAGGED1 expression and cleavage of J1-ICD. Silencing expression of JAGGED1 reduces IL-4-induced CLL viability, suggesting JAGGED1 promotes survival signals within CLL, although the molecular mechanism is unclear [102].

These studies demonstrate the importance of cytokines in mediating Notch ligand expression to promote survival of hematological malignancies. Further, Notch signaling supports survival of cancer cells through altered Notch expression. Understanding the role of Notch ligands in disease where Notch ligand-receptor interaction drives aberrant Notch activation may lead to important therapeutic targets [128].

## 8. Non-Canonical Cell-Autonomous Pathways

Regulating the spatial and temporal expression of Notch components governs Notch activation in many cell types. An alternative mechanism of Notch signaling through autonomous cis-interaction of receptor and ligand has been proposed, namely cis-inhibition (Figure 4C). In vitro cis-inhibition has been observed through time-lapse microscopy and mathematical modelling of NOTCH1 and Delta ligand. This study revealed dynamic switching between Notch activation and inhibition [129].

In vitro stimulation of CD4+ T cells with CD28 or CD3 regulates expression of Dll1, Dll4 and Jagged1. In turn, Notch ligands interact in cis with Notch receptors to repress Notch signaling within T cells, thereby regulating T cell activation. Exposing activated CD4+ T cells to soluble NOTCH1 ligand blocks cis-inhibition and in turn activates Notch signaling [130].

Pelullo and co-workers [131] utilized human T-ALL cell lines and a T-ALL mouse model to demonstrate that both Notch3 and Jagged1 can contribute to T-ALL. Notch3 activation induces Notch target genes including *Jagged1*. Further, cleaved J1-ICD interacts with the NIC3-CBF1-MAML1 transcriptional complex at a CBF1 binding site within the *Jagged1* proximal promoter, stimulating autocrine expression. Constitutive JAGGED1 cleavage is detected in multiple human T-ALL cell lines and knock down of NOTCH3 reduces *JAGGED1* transcriptional expression. Lastly, cleavage of the extracellular domain of Jagged1 into a soluble protein activates Notch signaling in surrounding cells [131]. Together, increased survival and migration of T-ALL in response to dysregulated JAGGED1 expression should be pursued as a potential new target for T-ALL treatment.

Notch signaling plays a pivotal role in epidermal stem cell homeostasis: self-renewal or differentiation. As epidermal stem cells begin to differentiate into keratinocytes, NOTCH1 is strongly expressed with levels gradually decreasing thereafter; an inverted pattern of expression is observed for NOTCH3. Both JAGGED2 and DLL1 show decreasing expression as epidermal stem cells differentiate. In human epidermal cell cultures, DLL1 was shown to cis-inhibit Notch receptors, blocking interaction with both JAGGED1 and JAGGED2 on neighboring cells, thus suppressing canonical Notch signaling and mediating epidermal stem cell differentiation [132].

While there are no definitive studies documenting Notch cis-inhibition in vivo, we note animal models that are consistent with a functional role for cis-inhibition. For example, during spermatogenesis in rat testis, Notch signaling is mediated by testosterone levels. Inhibition of testosterone-mediated transcriptional activation results in decreased Dll4 expression, with a concomitant increase in Dll1 and Jagged1 expression. Interestingly, Notch1 and canonical Notch target genes, Hes1 and Hey1, expression was suppressed, provoking speculation of cis-inhibition [133]. Furthermore, in a rat model of early mouse pregnancy prior to embryonic implantation in the uterus, the formation of decidua capillaries was cued by Notch signaling. Whilst pregnancies were unaffected in *Jagged1* knockout mutant mice, Dll4 expression increased in capillary endothelial cells. Concomitant increase in *Hey2* mRNA expression was observed, indicative of activated Notch signaling due to the loss of Jagged1 inhibition [134].

A recent study reporting cis-activation by Notch further illustrates the potential complexity of cell-autonomous regulation of Notch receptor-ligand interaction [135]. The conundrum of cis-inhibition in mammalian cells is enticing and whilst recent studies described above show many correlative relationships, this phenomenon requires further investigation [136,137].

## 9. Concluding Remarks

To be, or Notch to be, is a cell context dependent question whereby Notch signaling mediates cell fate. Here we present studies showing the importance of Notch signaling in mammalian development: it is patent that Notch signaling is multifaceted and crucial during embryogenesis. Similarly, Notch signaling is a gatekeeper of common lymphoid progenitor differentiation to either the T or B lineage through mediation of terminal differentiation and activation of both lymphoid lineages.

Mutations that disrupt Notch signaling during development are associated with a variety of rare hereditary diseases, hematological malignancies, and solid cancers. The capacity of Notch ligands to function as transcription co-factors, as well as influence surrounding cells, adds a layer of complexity to the Notch signaling repertoire. Further, the ability of Notch ligands to cis-interact with their cognate receptor expands the potential pathways for regulating Notch signaling. Clarity of Notch ligand function via bi-directional and lateral signaling in the tumor microenvironment and in promoting cancer cell survival will aid in designing therapeutic strategies to limit proliferation of oncogenic cells. With advancement in single cell sequencing technology and super resolution microscopy, unravelling the complexities of Notch signaling in regulating development and disease pathogenesis is becoming more achievable.

## Figures and Tables

**Figure 1 biomolecules-11-00849-f001:**
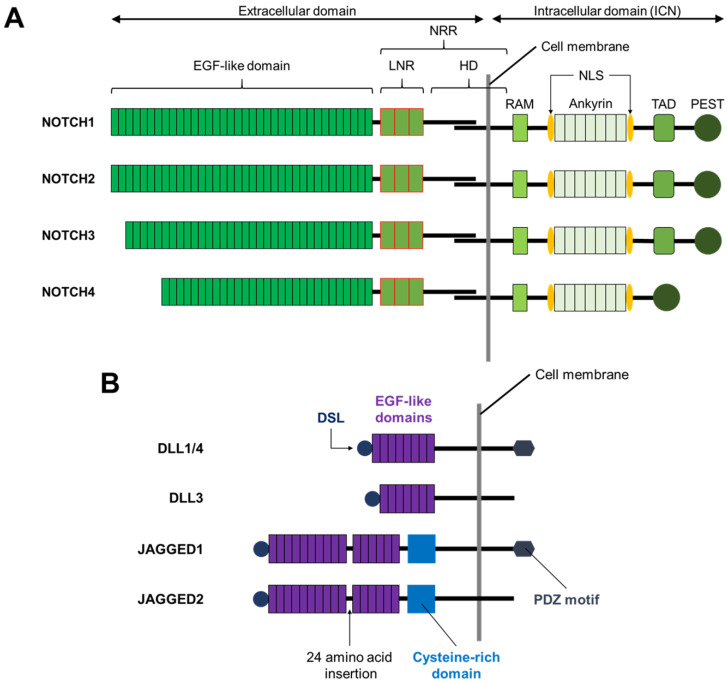
Structure of Notch receptors and ligands. (**A**) The four human Notch receptors (NOTCH1-4) are structurally similar. All receptors possess extracellular domains, and all four receptors share similarity in the intracellular domain, but NOTCH4 lacks the TAD domain. (**B**) All human Notch ligands of both the Delta-Like and Jagged families contain a DSL. The Notch ligand families can be distinguished by the number of EGF-like extracellular domains. Unique to the Jagged family is a 24 amino acid insertion between the 10th and 11th EGF-like domains of JAGGED1 and JAGGED2. The cysteine-rich region is another unique feature of the Jagged family ligands. At the C-terminus of DLL1, DLL4, and JAGGED1 is a PDZ motif. EGF: Epidermal growth factor; LNR: *lin-12/Notch* repeats; HD: Heterodimerization domain; NRR: Negative regulatory region; RAM: RBPjκ association module; NLS: Nuclear localization signal; TAD: Transcriptional activation domain; PEST: Proline (P), Glutamic acid (E), Serine (S), Threonine (T) rich domain; DLL: Delta-like; DSL: Delta/Serrate/LAG- 2 domain; PDZ: Post synaptic density protein (PSD95), Drosophila disc large tumor suppressor (Dlg1), and Zonula occludens-1 (ZO-1).

**Figure 2 biomolecules-11-00849-f002:**
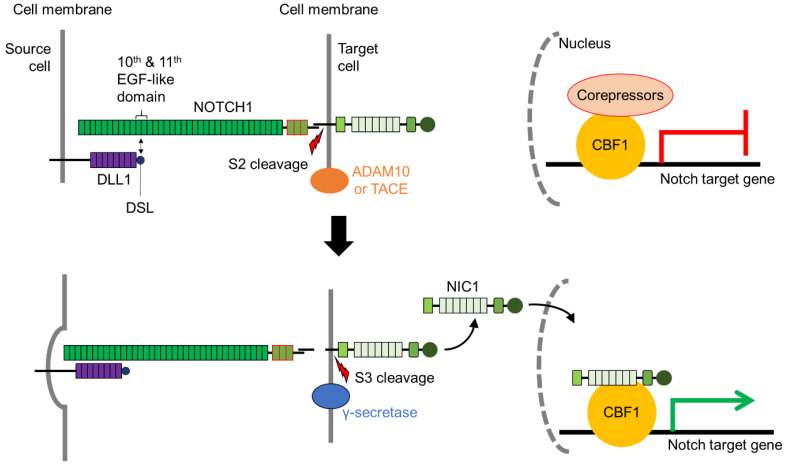
Trans-interaction of receptor and ligand activation of Notch signaling results in the transcription of CBF1-bound Notch target genes. Interaction of DELTA-LIKE1 (DLL1) DSL domain with the 10th and 11th EGF-like domain activates endocytosis by the source cell. This exposes the S2 site for cleavage by ADAM10 or TACE. Prior to Notch activation, Notch target genes are silenced by the recruitment of corepressors to CBF1. S2 cleavage exposes key amino acids to S3 cleavage by γ-secretase, releasing NIC1 into the nucleus of the target cell. NIC1 binds to CBF1 and displaces corepressors to upregulate Notch target gene expression. ADAM10: A disintegrin and metalloprotease 10; CBF1: C-promoter binding factor 1 (also termed RBPjκ); EGF-like: Epidermal growth factor-like; NIC1: Intracellular domain of NOTCH1; TACE: ADAM17/Tumor necrosis factor-α converting enzyme.

**Figure 3 biomolecules-11-00849-f003:**
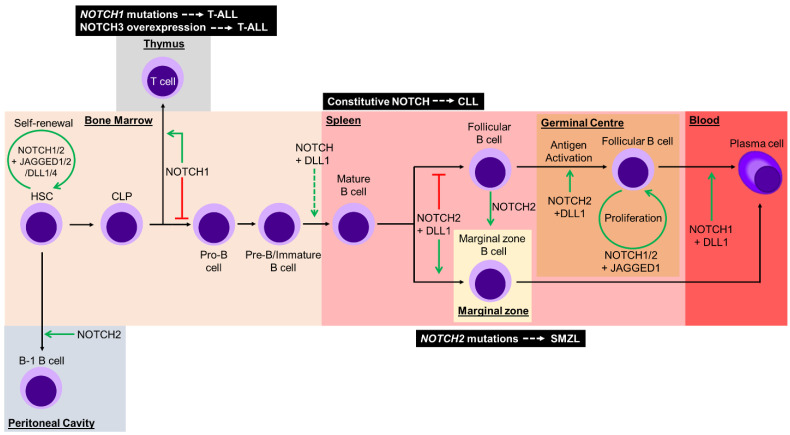
Development of hematopoietic stem cells (HSCs) from common lymphoid progenitors (CLPs) through to terminally differentiated plasma cells requires mediation of Notch signals at multiple stages. Within the bone marrow, HSCs are maintained in their pluripotent state through Notch signaling with surrounding stromal cells. NOTCH2 signaling promotes peritoneal B1 B cell differentiation. CLPs encounter a Notch signaling checkpoint that determines T or B cell fate through activation or inhibition of NOTCH1 signaling, respectively. Notch signaling with DELTA-LIKE1 (DLL1) is proposed to function as another checkpoint alongside negative selection against self-antigen of pre-B/immature B cells, prior to migration from the bone marrow into secondary lymphoid organs. Within the spleen, mature B cells develop into marginal zone B cells with NOTCH2 and DLL1 mediating Notch activation, whilst the absence of Notch signaling regulates follicular B cells. Notch signal is capable of mediating plasticity and inducing transdifferentiation from follicular B cells to functional marginal zone B cells. Upon antigen encounter in the germinal center, NOTCH2- and DLL1-mediated Notch signaling induces follicular B cell activation. Proliferation of activated follicular B cells is mediated by Notch activation through JAGGED1. Lastly, NOTCH1 and DLL1 mediated Notch signaling promotes terminal differentiation to plasma cells. Green arrows represent Notch signaling, whilst the red lines represent the absence of Notch signaling. The green dashed arrow represents a yet to be elucidated requirement for Notch signaling. Dysregulation of Notch signaling can contribute to lymphoid malignancies (black boxes) at different stages of development. T-ALL: T-acute lymphoblastic leukemia; CLL: chronic lymphocytic leukemia; SMZL: splenic marginal-zone lymphoma.

**Figure 4 biomolecules-11-00849-f004:**
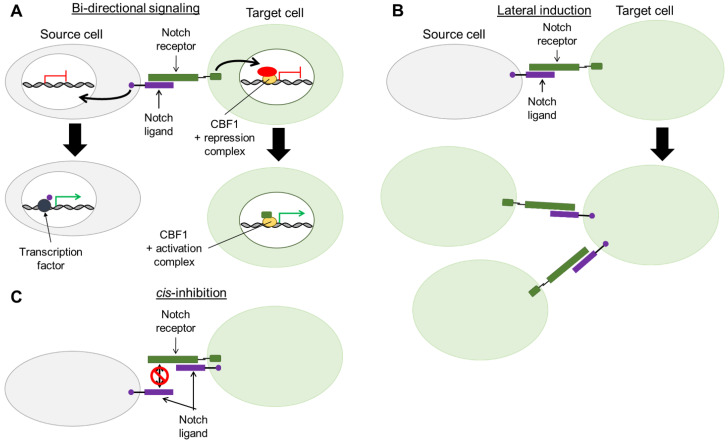
Mechanisms of Notch signaling regulation. (**A**) Following Notch activation, bi-directional Notch signaling can be initiated in both source and target cells. Cleavage of the intracellular domain of both Notch receptor and ligand can occur. These cleaved proteins translocate into the nucleus of their respective cells to function as transcription factors. (**B**) Notch activation in target cells can result in expression of Notch ligand. Such cells can subsequently function as source cells to activate Notch signaling in neighboring cells. This phenomenon is termed lateral induction. (**C**) Canonical Notch signaling can be blocked through cis-inhibition. This occurs when interaction of Notch receptor and ligand on the same target cell impedes the traditional trans-interaction of the ligand-expressing source cell with the target cell’s Notch receptor. CBF1: C-promoter binding factor 1 (also termed RBPjκ).

**Table 1 biomolecules-11-00849-t001:** Knockout mouse models of the Notch signaling pathway.

Knockout	Genotype	Embryonic Lethality	Developmental Impairment	References
***Notch1***	−/−	Yes (E9.5)	Somitogenesis Neurogenesis Developmental impairment	[24,25]
***Notch2***	*−/−*	Yes (E11.5)	Wide-spread pycnosis and apoptosis	[26]
***Notch3***	*−/−*	No	Vascular and arterial development	[27]
***Notch1 & Notch3***	*Notch1*: *−/−* *Notch3*: *−/−*	Yes (E9.5)	Somitogenesis Neurogenesis Developmental impairment	[28]
***Notch4***	*−/−*	No	Vasculature of embryonic head	[29,30]
***Notch1 & Notch4***	*Notch1*: *−/−* *Notch4*: *−/−*	Yes (E10.5)	Vascular development Developmental impairment (more severe than *Notch1* knockout)	[29]
***Dll1***	*−/−*	Yes (E10–E12)	Somitogenesis	[31]
***Dll3***	*−/−*	Yes (~20%)	Somitogenesis Skeletal deformities	[32]
***Dll4***	*−/−*	Yes (E8.5)	Vascular and arterial development	[33]
***Dll4***	+/−	Yes (E9.5–E10.5)	Vascular and arterial development	[33,34,35]
***Jagged1***	*−/−*	Yes (E10.5)	Vascular development	[36]
***Jagged2***	*−/−*	Yes (E10.5)	Craniofacial and skeletal deformities	[37,38]
***Cbf1***	*−/−*	Yes (E8.5)	Somitogenesis Placenta defects	[23]
***Adam10***	*−/−*	Yes (E9.5)	Somitogenesis Heart and neural defects	[17]
***Adam17/Tace***	*−/−*	Yes (E17.5)	Eyes and lungs defects	[16]

−/−: Homozygous Knockout; +/−: Heterozygous Knockout; Adam10: A disintegrin and metalloproteases 10; Cbf1: C-promoter binding factor 1; Dll: Delta-like; E: postimplantation days; Jag: Jagged; Tace: ADAM17/Tumor-necrosis-factor (TNF) α converting enzyme.

## Data Availability

Not applicable.

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
