# Peer review of "To Be, or Notch to Be: Mediating Cell Fate from Embryogenesis to Lymphopoiesis"

_biomolecules, 2021, doi:10.3390/biom11060849_

Round 1
Reviewer 1 Report
Ng and coworkers summarize in this detailed review article how Notch family regulates the biology of lymphoid cells with a complete introduction about the family member roles in embryogenesis. In the text, they describe different aspects ranging from the involvement in lymphoid cell physiology and homeostasis to the alterations of Notch pathway associated to pathology.
The review is well written and the sections follow a logical order. The figures and the table help the comprehension of Notch roles and effects. I only have some minor points to report.
As expected, the review is especially focused on Notch1 and Notch2 signalling and function, since these two receptors, as well explained in the paper, are probably the most effective among the four receptors of the family that work in a redundant way. As partially reported by the authors, few papers suggest that Notch3 and -4 also take part to immune system cell physiological and pathological processes. Therefore, it might be worth it to expand the paper with a few sentences describing the role of Notch3 and -4. The authors could consider:
Kamga, P.T. et al. Inhibition of Notch Signaling Enhances Chemosensitivity in B-cell Precursor Acute Lymphoblastic Leukemia. Cancer Res. 2019.
Bellavia, D. et al. Combined expression of pTα and Notch 3 in T cell leukemia identifies the requirement of preTCR for leukemogenesis. Proc. Natl. Acad. Sci. USA 2002.
Giuli, M.V. et al. Notch 3 contributes to T-cell leukemia growth via regulation of the unfolded protein response. Oncogenesis 2020.
Tottone, L. et al. Histone Modifications Drive Aberrant Notch 3 Expression/Activity and Growth in T-ALL. Front. Oncol. 2019.
Pinazza, M. et al. Histone deacetylase 6 controls Notch 3 trafficking and degradation in T-cell acute lymphoblastic leukemia cells. Oncogene 2018.
Ye, Q. et al. Expression of constitutively active Notch 4 (Int-3) modulates myeloid proliferation and differentiation and promotes expansion of hematopoietic progenitors. Leukemia 2004
Kamdje, A.H.N.; Bassi, G.S.; Pacelli, L.; Malpeli, G.; Amati, E.; Nichele, I.; Pizzolo, G.; Krampera, M. Role of stromal cell-mediated Notch signaling in CLL resistance to chemotherapy. Blood Cancer J. 2012.
Vercauteren, S.; Sutherland, H.J. Constitutively active Notch 4 promotes early human hematopoietic progenitor cell maintenance while inhibiting differentiation and causes lymphoid abnormalities in vivo. Blood 2004
Page 9, line 338-339. As the authors stated Notch signaling seems to be involved in the pathogenesis of HCV-related lymphomas and more in general, lymphoproliferative disorders. Together with somatic mutations described for Notch1 and Notch2, Notch4 germline SNPs were found associated in a benign pre-lymphomatous disorder named cryo-vasculitis, and in HCV-related lymphomas (Gragnani, L et al. Oncotarget 2017; Zignego, A.L. et al. Genes Immun. 2014; Genetic and clinical data predict onset of cryoglobulinemia in HCV patients and cryoglobulins clearance Artemova M. et al. Dig Liver Dis. 2018). The authors should consider these data.
Regarding to the sentence describing genetic mutations the authors could specify if there are germline or acquired (somatic).
A very few typos are present in the manuscript i.e.:
Page 8 line 269, please correct “sybdrome”, line 272 “… gain of function Notch signaling…” an of is missing between function and Notch.
Author Response
Response to Reviewer 1 Comments:
Point 1:
As expected, the review is especially focused on Notch1 and Notch2 signalling and function, since these two receptors, as well explained in the paper, are probably the most effective among the four receptors of the family that work in a redundant way. As partially reported by the authors, few papers suggest that Notch3 and -4 also take part to immune system cell physiological and pathological processes. Therefore, it might be worth it to expand the paper with a few sentences describing the role of Notch3 and -4.
Response:
We thank the reviewer for the suggested list of papers regarding Notch3 and Notch4. We have incorporated a summary of the papers in the following lines:
Lines 199-201: In transduced human cord cells, overexpression of NIC4 drives an immature CD4+ CD8+ T cell phenotype in bone marrow and spleen. These cells do not differentiate into B cells, suggesting these common lymphoid progenitors adopt a T cell fate [53].
Lines 309-315: NOTCH3 is a putative NOTCH1 transcriptional target and increased NOTCH3 surface expression together with hyperactivation is frequently observed in T-ALL [97] (Figure 3). Indeed, increased NOTCH3 expression promotes T-ALL survival during endoplasmic reticulum stress [98]. Further, NOTCH3 expression in T-ALL is post-translationally regulated through the lysosomal pathway, providing a potential therapeutic target via histone deacetylase inhibition [99].
Lines 424-428: Notch signaling between CLL and stromal cells promotes survival and chemoresistance to drug treatment. CLL cells co-cultured with bone marrow-derived mesenchymal stem cells display increased NOTCH4 and DLL3 expression. Further, in these co-cultures the mesenchymal stem cells also overexpress NOTCH4, suggestive of lateral induction of Notch signaling [125].
Point 2:
Page 9, line 344-349. As the authors stated Notch signaling seems to be involved in the pathogenesis of HCV-related lymphomas and more in general, lymphoproliferative disorders. Together with somatic mutations described for Notch1 and Notch2, Notch4 germline SNPs were found associated in a benign pre-lymphomatous disorder named cryo-vasculitis, and in HCV-related lymphomas (Gragnani, L et al. Oncotarget 2017; Zignego, A.L. et al. Genes Immun. 2014; Genetic and clinical data predict onset of cryoglobulinemia in HCV patients and cryoglobulins clearance Artemova M. et al. Dig Liver Dis. 2018). The authors should consider these data.
Response: Whilst we agree that the involvement of Notch signaling in the pathogenesis of HCV-related lymphomas is intriguing, the discussion of SNPs associated with these lymphoproliferative disorders is beyond the scope of this review.
Point 3:
A very few typos are present in the manuscript i.e.: Page 8 line 274, please correct “sybdrome”, line 277 “…gain of function Notch signaling…” and of is missing between function and Notch.
Response:
We apologize for the oversight of these errors and have corrected them:
Line 274: The typo for ‘syndrome’ has been corrected
Line 277: ‘of’ has been added between ‘gain of function’ and ‘Notch signaling’
Reviewer 2 Report
This is a nice review on Notch signaling highlighting the importance of Notch in development and disease. The review is clearly written and illustrated. Appropriate citations of classic and modern work is used. It would be useful for potential readers to address to more modern papers and reviews of Kovall, Kopan, Sprinzak and Bray groups than it is in the current version of the manuscript. There are some typos through the manuscript that should be corrected.
Author Response
Response to Reviewer 2 Comments:
Point 1:
It would be useful for potential readers to address to more modern papers and reviews of Kovall, Kopan, Sprinzak and Bray groups than it is in the current version of the manuscript.
Response:
We thank the reviewer for their positive comments and have updated the review references to include manuscripts from the Lendahl, Kopan, Kovall, Sprinzak, and Bray groups.
Refer to References 1, 3, 4, 5 and 22.
Point 2:
There are some typos through the manuscript that should be corrected.
Response:
We apologize for the oversight of these errors and have now corrected them within the document.
Reviewer 3 Report
The review gives a broad overview of the NOTCH research field, and it really starts "for beginners" - at the absolute basics of NOTCH signaling. One may wonder if this is still necessary to the extent done here. Most of this is now available on multiple even web pages, molecular signaling websites, etc. However, figures 1 and 2 that go with this are very clear and self-explanatory. I actually consider this one of the best overviews of the complex signaling mechanism I have seen so far. Also, the level at which the subsequent proteolytic steps (S2 and S3) occur, followed by transcriptional activation (or rather de-repression), warrants the inclusion of this part.
Similarly, the summary of embryonal effects for all of the possible loss of function mutations/knock-out is very detailed, but I have not seen it in this comprehensive form, which definitely provides a very good overview. Usually, the ligands are missing in such descriptive summaries.
Nevertheless, all of this serves as a lengthy (although acceptable) prelude to the main topic of this review - the role of NOTCH signaling in the hematopoietic system.
Also figure 3 is a very clear summary of the (known) signaling mechanisms involving NOTCH during the hematopoietic differentiation. Again, this is one of the most clear graphics to explain this process I can recollect. However, not everything is clear to anon-experts in hematopoiesis, for example: what are B-1 B-cells? It may also be of interest to indicate in Fig. 3 which role NOTCH1 and NOTCH2 mutations may play in leukemia (T-ALL and B-CLL) - along with these cell fate decisions? And how do NOTCH mutations support continued proliferation? That may be of interest to some readers. We are only returning to this area on page 9 of the review.
To me, the most interesting part of the entire review only starts at page 10 (Bi-directional NOTCH signaling) and the role of the microenvironment (page 10/11). Even more interesting (and novel) is the area we are starting at page 11 (non-canonical pathways); this is also where some speculation enters the scene - but it is well dosed. And we are also gradually leaving the original hematopoietic field.
These findings in the last few pages are rather complex and a bit more difficult to understand compared to the rest. I wonder if an additional table, or a graphical explanation, might be helpful there?
From there on, the review really delves deeply into the field of B-cell development and differentiation (of which I am not a great expert). But many highly relevant findings are referenced here, and this provides a very comprehensive overview of the entire field; usually utilizing more recent publications.
The survey of NOTCH-linked inherited diseases again provides a good overview of the most important of these diseases.
Altogether, I find this is a rather informative, and comprehensive review, nevertheless quite easy to read and understand. I wouldn't suggest many changes prior to publication.
Author Response
Response to Reviewer 3 Comments:
Point 1:
These findings in the last few pages are rather complex and a bit more difficult to understand compared to the rest. I wonder if an additional table, or a graphical explanation, might be helpful there?
Response:
We thank the reviewer for the comments regarding the latter sections of the manuscript. We have included a new graphical illustration to explain the concepts described therein. Figure 2B has been moved to Figure 4 (from lines 373 to 381) which in addition to cis-inhibition, also elucidates bi-directional and lateral induction of Notch signaling.
Point 2:
It may also be of interest to indicate in Fig. 3 which role NOTCH1 and NOTCH2 mutations may play in leukemia (T-ALL and B-CLL) - along with these cell fate decisions?
Response:
We have modified Figure 3 (Lines 167-183) to illustrate the contribution of Notch mutations to their respective hematological malignancies.